# The Association of Circulating Amino Acids and Dietary Inflammatory Potential with Muscle Health in Chinese Community-Dwelling Older People

**DOI:** 10.3390/nu14122471

**Published:** 2022-06-15

**Authors:** Yi Su, Amany Elshorbagy, Cheryl Turner, Helga Refsum, Timothy Kwok

**Affiliations:** 1Key Laboratory of Molecular Epidemiology of Hunan Province, School of Medicine, Hunan Normal University, Changsha 410013, China; alddle@hunnu.edu.cn; 2Department of Physiology, Faculty of Medicine, University of Alexandria, Alexandria 21526, Egypt; amany.elshorbagy@pharm.ox.ac.uk; 3Department of Pharmacology, University of Oxford, Oxford OX1 2JD, UK; cheryl.turner@pharm.ox.ac.uk; 4Institute of Basic Medical Sciences, Department of Nutrition, University of Oslo, 0316 Oslo, Norway; helga.refsum@medisin.uio.no; 5Department of Medicine and Therapeutics, Prince of Wales Hospital, The Chinese University of Hong Kong, Hong Kong 999077, China; 6Jockey Club Centre for Osteoporosis Care and Control, The Chinese University of Hong Kong, Hong Kong 999077, China

**Keywords:** amino acids, dietary inflammatory potential, muscle, aging

## Abstract

Amino acids (AAs) and dietary inflammatory potential play essential roles in muscle health. We examined the associations of dietary inflammatory index (DII) of habitual diet with serum AA profile, and ascertained if the associations between DII and muscle outcomes were mediated by serum AAs, in 2994 older Chinese community-dwelling men and women (mean age 72 years) in Hong Kong. Higher serum branched chain AAs (BCAAs), aromatic AAs and total glutathione (tGSH) were generally associated with better muscle status at baseline. A more pro-inflammatory diet, correlating with higher serum total homocysteine and cystathionine, was directly (90.2%) and indirectly (9.8%) through lower tGSH associated with 4-year decline in hand grip strength in men. Higher tGSH was associated with favorable 4-year changes in hand grip strength, gait speed and time needed for 5-time chair stands in men and 4-year change in muscle mass in women. Higher leucine and isoleucine were associated with decreased risk of sarcopenia in men; the associations were abolished after adjustment for BMI. In older men, perturbations in serum sulfur AAs metabolism may be biomarkers of DII related adverse muscle status, while the lower risk of sarcopenia with higher BCAAs may partly be due to preserved BMI.

## 1. Introduction

Degeneration of skeletal muscle leading to impaired function is a serious health problem in an aging society. The risk of adverse health outcomes, such as disability and mortality, increase with the decline in function of skeletal muscle in older people [1]. 

Amino acids (AA)s, with both their nutritional and regulatory functions, play an essential role in skeletal muscle metabolism and function [2]. Circulating branched-chain amino acid (BCAA; valine, leucine, and isoleucine) concentrations were positively associated with fat free mass [3] and muscle strength [4], and were positively incorporated in signaling pathways for muscle protein anabolism in healthy older adults [5]. BCAA supplements, especially those rich in leucine, were shown in randomized trials to enhance muscle protein synthesis [6], and improve muscle mass and function in older people [7]. These specific AAs and their active metabolites for muscle health have been endorsed by the European Society for Clinical Nutrition and Metabolism (ESPEN) expert group [8]. The sulfur containing AAs (SAAs), methionine and cysteine, are strictly required by skeletal muscle [9]. Glutathione, the antioxidant product of SAA metabolism, is closely related to the metabolic profile of skeletal muscle and could alter protein turnover in muscle [10]. Glutathione supplementation may improve acidification in skeletal muscles during exercise, leading to less muscle fatigue [11]. The supplementation of aromatic AAs, such as tryptophan, could potentially attenuate muscle loss caused by dietary protein deficiency [12].

Chronic inflammation plays a role in age-related skeletal muscle deterioration [13]. Elevated levels of circulating inflammatory markers were associated with accelerating loss in muscle mass and strength [14], and greater risk of sarcopenia in older people [15]. On the other hand, there is some evidence that chronic inflammation and AA metabolism are inter-related. For example, inflammation-associated markers of muscle are also associated with BCAA-related and tryptophan metabolic products in functionally limited older people [16]. Multicomponent exercises together with BCAA supplements increased serum albumin and reduced serum TNF-α in frail older people [17]. In addition, higher levels of serum inflammation biomarkers were associated with dietary intakes of SAAs [18]. 

A pro-inflammatory diet is defined as a diet that is associated with higher levels of circulating inflammatory markers, and is characterized by high content of fat and cholesterol and low content of fibre, omega-3 fatty acids, vitamin E, etc. [19]. Consuming a pro-inflammatory diet, as reflected by higher diet inflammatory index (DII) scores, has been shown to modulate inflammatory status [20] and also to influence muscle AA concentrations [21]. We have observed that higher DII scores were associated with loss of muscle strength and function in older men [22]. Whether these associations between dietary inflammatory potential and muscle health are mediated by circulating AAs is unknown and warrants investigation. 

We therefore examined the associations of DII with serum AA profile, and ascertained if the associations between DII and muscle outcomes were mediated by serum AAs, in a cohort of community-dwelling older Chinese people in Hong Kong. 

## 2. Materials and Methods

### 2.1. Study Participants

The evaluations were conducted in the study of Mr./Ms. OS (Hong Kong) cohort. This cohort included 2000 Chinese men and 2000 Chinese women aged ≥ 65 years, who were recruited from local communities from August 2001 to March 2003 [23]. Among these, 1424 men (71.2% of men) and 1573 women (78.7% of women) who were followed up by a visit to the research centre at year 4, had frozen serum available for assay of serum AAs. 

### 2.2. Demographic and Lifestyle Information

At baseline, demographics, lifestyle, and medical data were collected using a structured questionnaire administered by research assistants (RA). Physical Activity Scale of the Elderly (PASE) adapted for Hong Kong Chinese [24,25] was used to assess self-reported physical activity level. Body weight (kilograms) was measured using the Physician Beam Balance Scale (Healthometer, IL, USA) with the participants wearing an examination gown. Body height (centimeters) was measured using a Holtain Harpenden stadiometer (Holtain Ltd., Crosswell, UK). Body mass index (BMI) (kg/m^2^) was calculated.

### 2.3. Dietary Assessment and Dietary Inflammatory Index (DII)

The details of dietary assessment and DII calculation have been described previously [22]. Generally, daily and weekly intakes of 280 food items in the past year were assessed by RA at baseline using a locally developed and validated semi-quantitative food frequency questionnaire (FFQ) [26]. Mean nutrient quantitation and energy intakes per day were calculated mainly referring to food composition tables derived from the Chinese Medical Sciences Institute [9,13] and the Centre for Food Safety in Hong Kong [27], and supplementally referring to the food composition tables from Taiwan Food and Drug Administration [12] and the US Department of Agriculture (USDA) [28]. DII was developed by Dr. Hébert and et al., and the details have been provided elsewhere [19]. Briefly, a scoring algorithm for 45 food parameters, referring to their effects on the levels of inflammatory markers (IL-1β, IL-4, IL-6, IL-10, CRP, and TNF-α) was developed. In the present study, based on FFQ-derived dietary data, 30 out of 45 food parameters were available to calculate DII. The 30 food parameters were energy, protein, carbohydrate, total fat, saturated fat, monounsaturated fatty acid, polyunsaturated fatty acid, fibre, cholesterol, vitamin B12, vitamin B6, folic acid, thiamin, niacin, riboflavin, vitamin A, b-carotene, vitamin C, vitamin D, vitamin E, iron, magnesium, selenium, zinc, isoflavones, alcohol, caffeine, onion, pepper, and green/black tea. Baseline DII score was the sum of the transformed mean intake of each food composition parameters multiplied by their related inflammatory effect score [19]. A higher and positive DII score indicates a more pro-inflammatory diet, while a lower and negative one indicates a more anti-inflammatory diet. In the present study, a pro-inflammatory diet is defined as a diet in the highest tertile of DII in this community-dwelling older population [22].

### 2.4. AA Assays

Baseline serum samples were kept at −80 °C prior to analysis. AAs in serum were measured by liquid chromatography–tandem mass spectrometry (LC-MS/MS) using a modified version of a previously described method [29,30]. After pre-treatment, LC-MS/MS was carried out using a Shimadzu LC-20ADXR Prominence LC system (Kyoto, Japan) coupled to a Sciex QTRAP5500 mass spectrometer with a Turbo V ion source and TurboIonspray probe (Framingham, MA, USA). Chromatographic separation was achieved on a Phenomenex Kinetex Core Shell C18 (100 × 4.6 mm, 2.6 μm) LC column (Torrance, CA, USA) with an aqueous solution of formic acid [0.5%] and HFBA acid [0.3%] and acetonitrile gradient mobile phase. Positive mode multiple reaction monitoring was used for detection. Linear calibration curves of the peak area ratios of analyte and internal standard were used for quantification. Coefficient of variation (CV) for the analytes were 3–7%. Fifteen of the 19 analytes using spiked serum QA samples from an external quality assurance scheme ERNDIM (www.erndim.org, accessed on 7 June 2016) were used to validate the method. Measured serum AAs and their metabolites included BCAAs (valine, leucine, and isoleucine), aromatic AAs (phenylalanine, tryptophan, and tyrosine), and SAAs (methionine, total homocysteine (tHcy), cystathionine, total cysteine (tCys), taurine, and total glutathione (tGSH)). The Modification of Diet in Renal Disease study equation based on standardized serum creatinine, gender, and age was used to calculate the estimated glomerular filtration rate (eGFR) [31]. 

### 2.5. Assessment of Lean Muscle Mass

Lean muscle mass was measured by dual energy X-ray absorptiometry (DXA) using a Hologic QDR 4500 W device (Waltham, MA, USA). Measurement reproducibility was ensured by applying centralized quality control procedures, certificated DXA operators, and standardized procedures for scanning [32]. The CV of the scanners, estimated using a central phantom, was 0.8% for lean mass. Total appendicular lean muscle mass (ALM) was estimated as the sum of the appendicular lean mass minus bone mineral content of both arms and legs [33]. Height adjusted ALM (ALM/height2) was then calculated. Change of ALM was determined by the subtraction of ALM at baseline from that at year 4. 

### 2.6. Assessment of Muscle Strength and Physical Performance 

Muscle strength and physical performance were assessed at baseline and year 4. Hand grip strength was measured using a dynamometer (JAMAR Hand Dynamometer 5030JO; Sammons Preston, Bolingbrook, IL, USA). Two readings were taken from each side, and the maximum value was used for analysis. Participants were asked to walk at a normal pace twice along a six-metre-long straight line. The shorter time in completing the walking test was used to calculate the gait speed. The time (to the nearest decimal in seconds) taken to complete the 5-time chair stand test was recorded. Change of hand grip strength and time needed for 5-time chair stand test were determined by the subtraction of the value at baseline from that at year 4. 

Sarcopenia was defined as having both low lean mass and either low muscle strength or low physical performance, according to the consensus report proposed by the Asian Working Group for Sarcopenia (AWGS-2019) [34]. Gender-specific cut-off values of <7 kg/m^2^ for men and <5.4 kg/m_2_ for women were used to define low ALM. Low muscle strength was defined as having values of <28 kg for men and <18 kg for women on the hand grip test. Low physical performance was defined as having values of ≥12 s on the 5-time chair stand test for both genders or slow gait speed (<1.0 m/s). 

### 2.7. Statistical Analysis

Baseline characteristics, muscular measurements, and serum AAs among the DII category in tertiles were shown as mean (standard deviation, SD) in men and women respectively. Linear trends among DII categories were tested using analysis of variance (ANOVA) for continuous variables and Mantel–Haenszel test for categorical variables, and analysis of covariance (ANCOVA) and logistic regression model were used, respectively, to further adjust for age. Log-transformed values were used for variables with skewed distribution. 

General linear model was used to estimate the β coefficient with stand error (SE) of the category of DII score and AA concentrations in tertiles with the baseline or the 4-year changes of hand grip strength, gait speed, performance in chair stand test, and ALM, adjusting for age (or age and corresponding baseline measurement) in basic model A, or further adjusting for, BMI, eGFR, current smoking, physical activity level, previous fracture, hypertension, diabetes, chronic obstructive lung disease, cardiovascular disease, rheumatoid arthritis, nonsteroidal anti-inflammatory agent use, and osteoporosis medication in fully adjusted model B. Among those without sarcopenia at baseline, a binary logistic regression model with full adjustment (model B) was used to estimate the odds ratio (OR) with 95% confidence intervals (CI) for the risk of sarcopenia at year 4 per tertile of baseline DII or in AA concentration. Because circulating concentrations of several AAs are positively associated with BMI [35,36], and BMI largely influences risk of sarcopenia in older adults [37], the association between AAs and longitudinal change in muscle parameters and incident risk of sarcopenia were further investigated without adjusting for BMI in sub-fully adjusted model C. These analyses were conducted in men and women, separately. Mediation analysis [38] was conducted to explore whether serum AAs mediated the associations between the DII score and the longitudinal changes of muscular parameters. An estimate (c) of total effect (TE) of DII score and muscular parameters was estimated by a linear regression model 1 (Y = i1 + cX + e1). An estimate (c’) of direct effect (DE) and an estimate (ab) of indirect effect (IE) of DII on muscular parameters transmitted through AAs were estimated by the linear regression model 2 (M = i2 + aX + e2) and model 3 (Y = i3 + c′X + bM + e3). The standardized values of DII score, AA concentrations, and muscular parameters’ change were used in the mediation analysis for sensitivity test.

All statistical tests were two-tailed with *p* < 0.05 considered significant. To correct for multiple testing (of 12 individual AAs), Bonferroni adjustment was further used for AA-DII or AA-muscle parameters’ associations: a lower *p* value threshold of 0.004 (0.05/12) set for significance was further discussed. Statistical analyses were performed using SAS 9.4 (SAS Institute, Inc., Cary, NC, USA).

## 3. Results

1424 men and 1570 women (a total of 2994) had baseline serum AAs and dietary data, and muscular measurements at baseline and year 4, were included in the analysis. The mean (SD) age of the men and women was 71.7 (4.7) and 72.0 (5.1) years, respectively. Average DII score in men [−0.87 (SD = 1.3)] was lower than that in women [−0.22 (SD = 1.5), *p* < 0.001]. At baseline, 357 (25.1%) men and 176 (11.2%) women had sarcopenia. The average 4-year change of ALM, hand grip strength, gait speed, and time needed for 5-time stair test was −0.08 (SD = 0.32) kg/m_2_, −2.67 (SD = 4.78) kg, −0.10 (SD = 0.21) m/s, −2.55 (SD = 3.95) s in men, and −0.08 (SD = 0.30) kg/m_2_, −2.25 (SD = 3.49) kg, −0.10 (SD = 0.19) m/s, −1.45 (SD = 5.86) s in women, respectively. Among those without sarcopenia at baseline, 143 (13.4%) men and 112 (8.1%) women had sarcopenia at year 4.

### 3.1. Association between DII and Baseline Serum AAs

Table 1 shows participants’ baseline serum AA concentrations and muscular measurements by gender-stratified tertiles of DII score. Consuming a pro-inflammatory diet was significantly associated with higher serum isoleucine, tHcy and cystathionine concentrations in men (*p* < 0.050), but not in women. In the fully adjusted model B, only tGSH concentration in men was significantly and negatively associated with a higher DII category (*p* < 0.050). When a Bonferroni-adjusted *p* value threshold of significance was applied (*p* < 0.004), DII remained significantly associated with higher serum tHcy and cystathionine concentrations in men.

### 3.2. Associations of DII and Serum AAs with Muscle Parameters at Baseline

At baseline, men and women consuming a more pro-inflammatory diet had poorer performance in hand grip strength, gait speed, and 5-time chair stands test (*p* < 0.050). Men consuming a more pro-inflammatory diet also had less ALM (*p* < 0.050) and were more likely to have sarcopenia (*p* = 0.001) (Table 1). 

Higher BCAA and aromatic AA concentrations were generally associated with greater hand grip strength at baseline, with a fully adjusted β (SE) of 0.674 (0.213), 1.099 (0.214), 0.612 (0.214), and 0.547 (0.202) for each tertile increase in valine, leucine, isoleucine, and tryptophan respectively in men, and 0.278 (0.137), 0.575 (0.129), and 0.314 (0.132) for valine, tryptophan, and tyrosine in women. Higher concentrations of the SAAs tHcy (in both sexes), cystathionine (in women), and tCys (in men) were associated with poorer hand grip strength, whereas methionine concentration was associated with greater strength in men (Table 2). The valine, leucine, tryptophan, tHcy, cystathionine, and tCys associations remained significant after Bonferroni corrections (Table 2).

Higher BCAA, aromatic AA, and methionine concentrations were associated with greater gait speed in men but not women, whereas the SAAs tHcy (in men) and cystathionine and tGSH (in women) were associated with slower gait speed. The methionine and cystathionine associations remained significant after Bonferroni corrections. In the fully adjusted model B, each tertile increase in tHcy was significantly associated with 0.476 (0.133) s increase in the time needed to complete the 5-time chair stands test (*p* < 0.004; Table 2).

Higher SAAs tHcy, tCys, and taurine were associated with lower ALM in men and women, while cystathionine was further associated with lower ALM in men. The tHcy and tCys associations were significant after Bonferroni corrections. Neither the BCAAs nor the aromatic AAs were associated with ALM (Table 2).

### 3.3. Association of Baseline DII and Serum AAs with Change in Muscle Parameters over 4 Years

In longitudinal analysis, a higher DII score was associated with an accelerated loss of hand grip strength and gait speed at year 4 in men but not women. In the fully adjusted model B, each tertile increase in DII category was significantly associated with 0.321 (0.143) kg loss of grip strength and 0.014 (0.006) m/s loss of gait speed over 4 years in men (Table 3). Among those without sarcopenia at baseline, DII score was not significantly associated with the incident risk of sarcopenia at year 4 in both genders.

In men, higher tGSH concentration was associated with less loss in hand grip strength and gait speed, and less increase in time needed for 5-time chair stands in the fully adjusted model B (Table 3). In women, higher tGSH was associated with less ALM loss (β (SE): 0.028 (0.009), *p* < 0.004; Table 3). Higher cystathionine in men was associated with accelerated loss in hand grip strength, with a 0.310 kg decline in hand grip strength per tertile increase in cystathionine (*p* <0.050), and higher tCys correlated with an increase in time to complete the chair stands test (*p* < 0.050; Table 3). No consistent associations were observed for the other AAs with change in muscle parameters in men or women. According to the examinations in the fully adjusted model B and the sub-fully adjusted model C, inclusion of BMI as a covariate did not influence the associations between AAs and the 4-year change of individual muscular parameters in either gender (Table 3 and Appendix A).

Circulating concentrations of the BCAAs leucine and isoleucine, and the AAA tyrosine, were associated with a significant 20–25% decreased risk of developing sarcopenia at year 4 in men after adjustment for age, renal function, lifestyle, and chronic diseases in the sub-fully adjusted model C; a similar trend was observed in women but was not statistically significant (Table 3). No other AAs were associated with sarcopenia risk in the model C. The associations of leucine, isoleucine, and tyrosine with sarcopenia in men became non-significant after further adjustment for BMI in the fully adjusted model B (Table 3). In women, adjustment for BMI also revealed a significant positive association between leucine, tyrosine, methionine, taurine and risk of sarcopenia [OR (95%CI): 1.39 (1.05, 1.84), 1.35 (1.03, 1.77), 1.44 (1.10, 1.89), and 1.41 (1.08, 1.84), respectively; *p* < 0.050; Table 3].

### 3.4. Mediation Analysis

Since tGSH was the only metabolite that remained significantly associated with 4-year change in muscle function after Bonferroni correction in men, mediation analysis was further conducted to examine whether the observed associations between DII and the longitudinal changes in hand grip strength or gait speed in men (Table 3) are mediated by tGSH. In the fully adjusted model B, a significant direct effect (DE) of DII on loss in hand grip strength (c = −0.174) was observed, with a significant indirect effect (IE) through serum tGSH (ab = −0.055 × 0.311 = −0.017). The IE relative to the total effect (TE) was 9.8% (Figure 1A). A significant DE of DII on loss in gait speed (c = −0.010) was observed, but there was no significant mediation effect of serum tGSH (ab = −0.057 × 0.006 = −0.003). The IE relative to the TE was 3.4% (Figure 1B). The use of standardized values in the sensitivity analysis did not essentially alter the results. 

## 4. Discussion

In the present study of 2994 community-dwelling older men and women, consumption of a pro-inflammatory diet was associated with poorer muscle strength and function. In men but not women, a pro-inflammatory diet was also associated with serum concentrations of the SAAs tHcy and cystathionine, and with accelerated loss in muscle strength and function over 4 years. At baseline, higher concentrations of individual BCAAs and aromatic AAs were generally associated with greater muscle strength, whereas the SAAs tHcy, cystathionine, and tCys were associated with lower muscle mass and strength. Higher serum tGSH was associated with slower decline in muscle strength and function over 4 years in men, and appeared to mediate approximately 10% of the association of consuming a pro-inflammatory diet with decline in hand grip strength. Together, these data suggest that BCAAs and tGSH are linked to healthier muscle status in the older people, whereas consuming a pro-inflammatory diet and having higher concentrations of the SAAs tHcy, cystathionine, and tCys are associated with poorer muscle status. In addition, among those without sarcopenia at the beginning, BMI may confound the associations between specific AAs and the sarcopenia risk at year 4.

Our data shows that serum concentrations of individual AAs can be considered biomarkers of muscle health in the older people, although, with the exception of tGSH, they do not predict the rate of decline in muscle status. BCAAs and aromatic AAs were generally associated with better muscle status in the present study, which is consistent with the positive relationship between BCAA concentrations and muscle function observed in older Japanese women [39]. A landmark interventional study has established that increasing the proportion of leucine in an ingested mixture of essential AAs can reverse the age-related attenuated response of muscle protein synthesis in older subjects [40]. The increase in protein synthesis in response to leucine was proposed to be via activating mammalian target of rapamycin (mTOR) [41]. BCAAs supplementation was also found to attenuate muscle damage and ameliorate muscle soreness after resistance exercise in trained men [42]. 

On the other hand, specific SAAs, namely tHcy, cystathionine, and tCys were associated with poorer muscle status in the present study. To our knowledge, this is the first study to systematically investigate all SAAs in relation to muscle status in the older people. Our finding that serum cystathionine and tCys were associated with adverse muscle parameters in older subjects is novel, and might seem counterintuitive, since they are on the metabolic pathway from the essential SAA methionine to the key antioxidant tGSH. However, both cystathionine and tCys are emerging as biomarkers of adverse health in older subjects, including predicting type 2 diabetes [43], cardiovascular events [44], and mortality [45]. In the present study these SAAs also correlated with dietary inflammatory potential. Interestingly, deficient levels of reduced glutathione have been previously observed in subjects with elevated cystathionine, indicating the presence of oxidative stress [46]. Plasma cystathionine also correlates with circulating levels of the inflammatory marker C-reactive protein [47]. Earlier studies in animals have linked activation of cystathionine beta synthase enzyme, with subsequent upregulation of cystathionine and cysteine synthesis, to combating chronic inflammation states and increased oxidative stress [48]. Plasma tCys and cystathionine have also been associated in multiple cohorts with increased adiposity and dyslipidemia [36,43], which are known to be associated with chronic inflammation. Our data suggest that a pro-inflammatory diet, particularly in older men, may be one factor driving an unhealthy elevation of cystathionine and tCys in older subjects, with adverse consequences for muscle health. The more detrimental effect of pro-inflammatory diet consumption on musculoskeletal health in older men has been reported and discussed previously [22,49]. This gender difference in the influence of pro-inflammatory foods on chronic inflammation, alternations in AA metabolism, and decline of muscular health warrants further investigations.

Plasma tHcy, a recognized marker of oxidative stress [50], was investigated extensively in relation to muscle function, often in smaller cohorts compared with the present study, and often with mixed or null results [51]. As one of the pro-oxidant metabolites of methionine, higher homocysteine in the present study was related to lower muscle strength, in line with previous reports in older people in the Baltimore study of ageing [52]. Defective homocysteine metabolism has been proposed to impair the autonomic regulation of skeletal muscle blood vessels and hence affect long-term muscle function [53]. In the present study, we observed that elevated tHcy may additionally be a marker of a pro-inflammatory diet, which is in turn associated with poor muscle health. That circulating tGSH concentrations mediated some of the association between DII and decline in muscle strength in the present study further supports the concept that perturbations in the SAA metabolic pathway may be a biomarker of inflammatory or oxidative conditions associated with adverse health conditions in older subjects.

The association between circulating AAs and sarcopenia in older subjects has yielded inconsistent results in the literature. For example, plasma leucine and isoleucine were found in a cross-sectional analysis to be increased in individuals with low muscle quality relative to age-, sex-, and height-matched individuals with high muscle quality [54]. The opposite was found in a larger cohort, where sarcopenic individuals had decreased leucine and isoleucine concentrations [55]. These discrepancies are likely to be partly caused by the cross-sectional nature of these analyses, whereby circulating AAs influence the likelihood of sarcopenia, but a low muscle mass also alters circulating AAs; thus, the net association of AAs with sarcopenia will vary according to the stage of sarcopenia. Hence, prospective data are needed to determine whether circulating AAs influence the risk of sarcopenia in elderly subjects. Our prospective analysis showed that leucine and isoleucine were associated with lower risk of developing sarcopenia at year 4. Because BCAAs [35] are positively associated with BMI, and a higher BMI reduces the risk of sarcopenia in older adults [37], we tested whether adjusting for BMI altered these associations. Adjusting for BMI weakened the protective associations of BCAAs with sarcopenia risk (in men) and revealed increased risk of sarcopenia in association with some AAs in women. Collectively, our findings suggest that a large part of the apparent protective effect of higher BCAAs against developing sarcopenia is linked to the positive associations between BCAAs and BMI.

The strengths of this study include using a validated assessment of habitual dietary inflammatory potential, fasting measurements of a full serum amino acid profile, together with longitudinal follow-up for gold-standard DXA measures of muscle mass, as well as muscle functional parameters in a free-living population susceptible to age-related sarcopenia. Though, compared with the longitudinal measurements, the association of some AAs with incident sarcopenia risk were less obvious and might be confounded by BMI, which need further exploration. Data on key lifestyle and health-related confounders was available for adjustment in the analysis. However, with the exception of GSH, no metabolites of the AAs were measured, precluding mechanistically relevant pathway analyses. Limitations related to the possibility of recall bias in FFQ should also be noted. Further, dietary data and AA measurements were only available at baseline, thus it was not possible to investigate whether changes in these parameters correlated with change in muscle mass and function over 4 years. Finally, the study only included participants at a single-centre study in Hong Kong who attended an assessment at baseline and after years of follow-up; the findings therefore may not be generalizable to ethnic Chinese older individuals elsewhere.

## 5. Conclusions

In summary, consuming a diet with high inflammatory potential was associated with higher circulating concentrations of serum tHcy and cystathionine and low concentration of tGSH, and with pooper muscle function in Chinese community-dwelling older men. Serum BCAAs, aromatic AAs, and tGSH were generally associated with better muscle performance, whereas the tHcy, cystathionine, and tCys were associated with unfavorable muscle status, after adjustment for renal function and multiple comorbidities. Higher tGSH at baseline predicted slower decline in muscle mass in women and muscle function in men, and mediated a small fraction of the association of DII with decline in muscle strength in men. Overall, our data suggest that dietary habits that reduce oxidation or inflammation may be beneficial to muscle health in older people, and that perturbations in specific AA metabolic pathways may be a biomarker of dietary inflammatory potential associated with adverse muscle status. 

## Figures and Tables

**Figure 1 nutrients-14-02471-f001:**
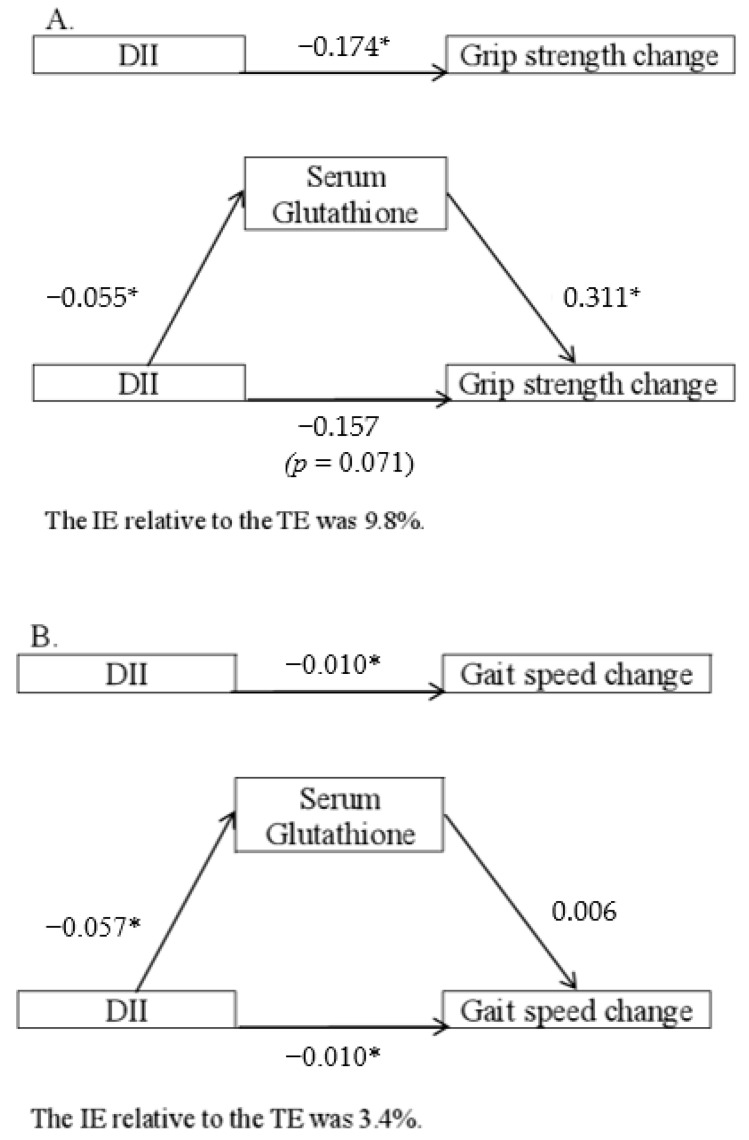
The mediation role of serum glutathione in the relationship between baseline dietary inflammatory index (DII) and the longitudinal changes of (**A**) hand grip strength and (**B**) gait speed (value at year 4 subtracted that at baseline). Notes: the β was adjusted for corresponding baseline measurement, body mass index, estimated glomerular filtration rate, current smoking, physical activity level, previous fracture, hypertension, diabetes, chronic obstructive lung disease, cardiovascular disease, rheumatoid arthritis, nonsteroidal anti-inflammatory agent use, and osteoporosis medication in the fully adjusted models. * *p* < 0.05.

**Table 1 nutrients-14-02471-t001:** The differences in muscular parameters and serum amino acids level of men and women among DII tertiles.

	Men, Mean (SD)					Women, Mean (SD)				
Variables		DII		*p*-Value ^a^	*p*-Value ^b^		DII		*p*-Value ^a^	*p*-Value ^b^
	T1, *n* = 498	T2, *n* = 475	T3, *n* = 451			T1, *n* = 552	T2, *n* = 522	T3, *n* = 496		
Age (year)	71.6 (4.8)	71.5 (4.5)	72.1 (4.8)	0.115	-	71.5 (4.9)	72.2 (5)	72.5 (5.3)	0.001	-
BMI (kg/m^2^)	23.6 (3.1)	23.6 (3)	23.3 (3.2)	0.116	0.154	24.0 (3.5)	24.1 (3.3)	23.8 (3.3)	0.474	0.636
Appendicular muscle mass (kg/m^2^)	7.3 (0.8)	7.3 (0.8)	7.2 (0.8)	0.015	0.029	6.1 (0.8)	6.1 (0.7)	6.1 (0.7)	0.105	0.173
Grip strength (kg)	35.2 (6.7)	34.8 (6.3)	33.8 (6.5)	0.001	0.003	23.0 (4.5)	22.4 (4.3)	22.2 (4.3)	0.003	0.029
Gait speed (m/s)	1.1 (0.2)	1.1 (0.2)	1.0 (0.2)	<0.001	<0.001	1.0 (0.2)	1.0 (0.2)	0.9 (0.2)	<0.001	0.001
5-time chair stand (s)	12.2 (3.6)	12.3 (3.4)	12.8 (3.8)	0.006	0.010	12.6 (4.5)	13.2 (4.7)	13.8 (5.4)	<0.001	0.001
Sarcopenia (yes, %)	111 (22.3%)	101 (21.3%)	145 (32.2%)	0.001	0.001	56 (10.1%)	63 (12.1%)	57 (11.5%)	0.476	0.653
Branched chain amino acids										
Valine (uM)	301.6 (45.6)	307.6 (45.1)	306.1 (44.4)	0.116	0.076	279 (44.4)	279.2 (42.7)	279.7 (45.9)	0.780	0.644
Leucine (uM)	155.8 (23.9)	158.6 (24.5)	158.1 (25.1)	0.137	0.089	140.8 (23)	140.5 (20.9)	140.3 (23.1)	0.699	0.953
Isoleucine (uM)	83.9 (14.8)	85.3 (15.4)	85.7 (14.7)	0.067	0.044	75.4 (13.7)	76.1 (13.8)	76 (14)	0.474	0.380
Aromatic amino acids										
Phenylalanine (uM)	97.7 (12.9)	98.3 (13.5)	98.8 (13.3)	0.180	0.188	96.7 (14.6)	96.7 (13.7)	96 (14.2)	0.471	0.405
Tryptophan (uM)	61.6 (10.5)	61.1 (10.7)	61.1 (11.9)	0.487	0.620	56.7 (10.1)	56 (9.7)	57 (11.6)	0.679	0.489
Tyrosine (uM)	85.1 (14.6)	84.5 (14.7)	83.6 (14.4)	0.108	0.122	82.6 (14.2)	82.9 (15)	81.8 (14.3)	0.420	0.380
Sulfur amino acids										
Methionine (uM)	31.2 (4.9)	31 (4.6)	30.9 (4.7)	0.314	0.389	23.9 (4.2)	24.1 (4.7)	23.7 (4.3)	0.525	0.571
tHcy (uM) ^c^	14.9 (4.2)	15.5 (4.8)	16.2 (6.6)	0.001	0.001	13 (4.5)	13.1 (4.3)	13.2 (3.9)	0.278 ^c^	0.853 ^c^
Cystathionine (nM) ^c^	278.3 (178.2)	280.6 (204.7)	303.8 (191.2)	0.003	0.0038	235.5 (262.5)	240.5 (150.4)	233.2 (140.4)	0.378 ^c^	0.745 ^c^
tCys (uM)	359.1 (44.6)	361.4 (40.6)	362.4 (49.3)	0.257	0.461	352.1 (46.6)	350.2 (42.4)	354.6 (43.4)	0.384	0.928
Taurine (uM)	170.0 (31.2)	172.3 (34.8)	172.8 (31.7)	0.158 ^c^	0.107 ^c^	171.7 (30.9)	170.7 (36)	169.5 (31)	0.235 ^c^	0.308 ^c^
tGSH (uM)	4.2 (1.4)	4.2 (1.4)	4.0 (1.2)	0.065	0.079	4.3 (1.4)	4.2 (1.3)	4.2 (1.3)	0.152	0.171

BMI: body mass index; DII: dietary inflammatory index; SD: standard deviation; tHcy: total homocysteine; tCys; total cysteine; tGSH: total glutathione. ^a^ *p* for linear trend was tested by analysis of variance (ANOVA) for continuous variables and by Mantel–Haenszel for categorical variables. ^b^ adjusted for age. ^c^ Using log-transformed value.

**Table 2 nutrients-14-02471-t002:** The association of DII and serum amino acids with the muscular parameters at baseline.

Amino Acids	Grip Strength (kg)	Gait Speed (m/s)	Chair Stand Test (s)	Appendicular Muscle Mass (kg/m^2^)
	Men			
	β Estimate (SE) ^a^	β Estimate (SE) ^a^	β Estimate (SE) ^a^	β Estimate (SE) ^a^
DII (per one tertile)	−0.557 (0.199) ^b^	−0.034 (0.007) ^b,c^	0.281 (0.117) ^b^	−0.024 (0.015)
Branched chain amino acids				
Valine (per one tertile)	0.674 (0.213) ^b,c^	0.015 (0.007) ^b^	−0.021 (0.126)	−0.025 (0.016)
Leucine (per one tertile)	1.099 (0.214) ^b,c^	0.010 (0.007)	−0.061 (0.127)	−0.011 (0.016)
Isoleucine (per one tertile)	0.612 (0.214) ^b^	0.009 (0.007)	−0.041 (0.126)	−0.043 (0.016) ^b^
Aromatic amino acids				
Phenylalanine (per one tertile)	0.069 (0.212)	0.011 (0.007)	−0.18 (0.124)	−0.023 (0.016)
Tryptophan (per one tertile)	0.547 (0.202) ^b^	0.007 (0.007)	−0.104 (0.119)	0.028 (0.015)
Tyrosine (per one tertile)	−0.391 (0.206)	0.019 (0.007) ^b^	−0.068 (0.121)	0 (0.016)
Sulfur amino acids				
Methionine (per one tertile)	0.541 (0.201) ^b^	0.025 (0.007) ^b,c^	−0.213 (0.118)	0.043 (0.015) ^b^
tHcy (per one tertile)	−0.873 (0.226) ^b,c^	−0.019 (0.008) ^b^	0.476 (0.133) ^b,c^	−0.073 (0.017) ^b,c^
Cystathionine (per one tertile)	−0.341 (0.212)	0.006 (0.007)	0.143 (0.125)	−0.039 (0.016) ^b^
tCys (per one tertile)	−0.636 (0.22) ^b,c^	−0.008 (0.008)	0.198 (0.129)	−0.06 (0.017) ^b,c^
Taurine (per one tertile)	−0.082 (0.205)	0.004 (0.007)	−0.188 (0.121)	−0.032 (0.015) ^b^
tGSH (per one tertile)	0.328 (0.203)	−0.009 (0.007)	0.226 (0.119)	0.031 (0.015) ^b^
		**Women**		
	**β Estimate (SE) ^a^**	**β Estimate (SE) ^a^**	**β Estimate (SE) ^a^**	**β Estimate (SE) ^a^**
DII (per one tertile)	−0.219 (0.129)	−0.015 (0.006) ^b^	0.442 (0.150) ^b,c^	−0.017 (0.014)
Branched chain amino acids				
Valine (per one tertile)	0.278 (0.137) ^b^	0.001 (0.006)	−0.054 (0.16)	−0.014 (0.015)
Leucine (per one tertile)	0.266 (0.138)	−0.002 (0.006)	−0.041 (0.161)	−0.014 (0.015)
Isoleucine (per one tertile)	0.321 (0.138)	0.004 (0.006)	−0.173 (0.161)	−0.011 (0.015)
Aromatic amino acids				
Phenylalanine (per one tertile)	−0.008 (0.134)	−0.005 (0.006)	0.098 (0.155)	−0.064 (0.015) ^b,c^
Tryptophan (per one tertile)	0.575 (0.129) ^b,c^	0.001 (0.006)	0.003 (0.15)	0.025 (0.014)
Tyrosine (per one tertile)	0.314 (0.132) ^b^	0.002 (0.006)	0.057 (0.154)	−0.019 (0.015)
Sulfur amino acids				
Methionine (per one tertile)	0.108 (0.13)	0.010 (0.006)	0.025 (0.151)	0.003 (0.015)
tHcy (per one tertile)	−0.343 (0.151) ^b^	−0.011 (0.007)	0.007 (0.176)	−0.054 (0.017) ^b,c^
Cystathionine (per one tertile)	−0.420 (0.141) ^b,c^	−0.022 (0.007) ^b,c^	0.227 (0.165)	−0.003 (0.016)
tCys (per one tertile)	0.020 (0.145)	−0.008 (0.007)	−0.162 (0.169)	−0.057 (0.016) ^b,c^
Taurine (per one tertile)	−0.129 (0.13)	−0.002 (0.006)	−0.059 (0.152)	−0.074 (0.014) ^b,c^
tGSH (per one tertile)	0.033 (0.131)	−0.012 (0.006) ^b^	0.106 (0.152)	0.010 (0.015)

DII: dietary inflammatory index; SE: standardized error. tHcy: total homocysteine; tCys; total cysteine; tGSH: total glutathione. ^a^ in fully adjusted model B: adjusted for baseline age, body mass index, estimated glomerular filtration rate, current smoking, physical activity level, previous fracture, hypertension, diabetes, chronic obstructive lung disease, cardiovascular disease, rheumatoid arthritis, nonsteroidal anti-inflammatory agent use and osteoporosis medication. ^b^ indicates for significant value (*p* < 0.05). ^c^ indicates for significant value (*p* < 0.004) for amino acids.

**Table 3 nutrients-14-02471-t003:** The association of baseline serum amino acids with the changes of individual muscular parameters and incident sarcopenia over 4 years.

AAs	Grip Strength (kg)	Gait Speed (m/s)	Chair Stand Test (s)	Appendicular Muscle Mass (kg/m^2^)	Sarcopenia ^d^ (yes)	Sarcopenia ^d^ (yes)
			Men			
	β Estimate (SE) ^a^	β Estimate (SE) ^a^	β Estimate (SE) ^a^	β Estimate (SE) ^a^	OR (95%CI) ^a^	OR (95%CI) ^e^
DII (per one tertile)	−0.321 (0.143) ^b^	−0.014 (0.006) ^b^	0.076 (0.104)	0.009 (0.01)	1.09 (0.86, 1.40)	1.12 (0.89, 1.41)
Branched chain amino acids						
Valine (per one tertile)	−0.073 (0.154)	0 (0.007)	−0.014 (0.111)	−0.016 (0.011)	1.10 (0.84, 1.43)	0.79 (0.62, 1.00)
Leucine (per one tertile)	−0.046 (0.156)	0.004 (0.007)	−0.178 (0.112)	0.001 (0.011)	0.94 (0.72, 1.22)	0.70 (0.55, 0.90) ^b^
Isoleucine (per one tertile)	0.077 (0.154)	0.001 (0.007)	−0.131 (0.112)	−0.006 (0.011)	0.96 (0.74, 1.24)	0.74 (0.58, 0.95) ^b^
Aromatic amino acids						
Phenylalanine (per one tertile)	−0.045 (0.151)	0.001 (0.006)	−0.146 (0.11)	0.001 (0.011)	1.01 (0.78, 1.31)	0.82 (0.65, 1.04)
Tryptophan (per one tertile)	0.168 (0.145)	0.012 (0.006)	0.042 (0.105)	0.013 (0.01)	1.09 (0.85, 1.39)	0.99 (0.79, 1.25)
Tyrosine (per one tertile)	0.317 (0.148) ^b^	0 (0.006)	−0.074 (0.108)	0.003 (0.011)	0.97 (0.75, 1.24)	0.75 (0.60, 0.95) ^b^
Sulfur amino acids						
Methionine (per one tertile)	0.203 (0.144)	0.001 (0.006)	−0.195 (0.105)	−0.004 (0.01)	0.88 (0.69, 1.13)	0.88 (0.69, 1.10)
tHcy (per one tertile)	−0.016 (0.163)	0.002 (0.007)	0.199 (0.118)	−0.012 (0.012)	0.97 (0.73, 1.27)	0.84 (0.65, 1.09)
Cystathionine (per one tertile)	−0.310 (0.152) ^b^	0.002 (0.006)	−0.158 (0.11)	−0.016 (0.011)	1.10 (0.84, 1.44)	0.98 (0.76, 1.25)
tCys (per one tertile)	−0.265 (0.158)	0.004 (0.007)	0.266 (0.114) ^b^	−0.008 (0.011)	1.42 (1.08, 1.88) ^b^	1.26 (0.98, 1.63)
Taurine (per one tertile)	−0.007 (0.147)	0.010 (0.006)	−0.198 (0.106)	−0.013 (0.011)	1.01 (0.79, 1.30)	1.07 (0.85, 1.36)
tGSH (per one tertile)	0.483 (0.145) ^b,c^	0.014 (0.006) ^b^	−0.260 (0.105) ^b^	0.006 (0.010)	0.88 (0.68, 1.13)	1.00 (0.79, 1.27)
			**Women**			
	**β estimate (SE) ^a^**	**β estimate (SE) ^a^**	**β estimate (SE) ^a^**	**β estimate (SE) ^a^**	**OR (95%CI) ^a^**	**OR (95%CI) ^e^**
DII (per one tertile)	0.070 (0.099)	−0.004 (0.005)	0.182 (0.173)	−0.007 (0.009)	0.82 (0.63, 1.06)	0.80 (0.63, 1.02)
Branched chain amino acids						
Valine (per one tertile)	0.02 (0.106)	0.006 (0.006)	−0.228 (0.184)	−0.019 (0.01)	1.27 (0.96, 1.67)	0.78 (0.61, 1.01)
Leucine (per one tertile)	0.069 (0.107)	0.003 (0.006)	−0.235 (0.186)	−0.009 (0.01)	1.39 (1.05, 1.84) ^b^	0.88 (0.68, 1.13)
Isoleucine (per one tertile)	−0.045 (0.107)	−0.001 (0.006)	0.1 (0.185)	−0.017 (0.01)	1.30 (0.99, 1.71)	0.83 (0.65, 1.07)
Aromatic amino acids						
Phenylalanine (per one tertile)	0.135 (0.103)	0.009 (0.006)	−0.156 (0.178)	0.006 (0.01)	1.22 (0.93, 1.60)	0.85 (0.66, 1.09)
Tryptophan (per one tertile)	−0.022 (0.100)	0.006 (0.005)	−0.361 (0.173) ^b^	−0.023 (0.009) ^b^	1.22 (0.94, 1.58)	1.00 (0.78, 1.27)
Tyrosine (per one tertile)	0.052 (0.102)	0.002 (0.005)	−0.271 (0.177)	−0.011 (0.01)	1.35 (1.03, 1.77) ^b^	0.94 (0.74, 1.20)
Sulfur amino acids						
Methionine (per one tertile)	0.16 (0.100)	−0.007 (0.005)	−0.007 (0.174)	−0.007 (0.009)	1.44 (1.10, 1.89) ^b^	1.10 (0.86, 1.41)
tHcy (per one tertile)	−0.107 (0.117)	−0.003 (0.006)	−0.062 (0.203)	0.005 (0.011)	1.02 (0.75, 1.37)	0.79 (0.59, 1.04)
Cystathionine (per one tertile)	−0.129 (0.109)	−0.002 (0.006)	−0.022 (0.189)	−0.006 (0.010)	0.96 (0.72, 1.28)	0.78 (0.60, 1.02)
tCys (per one tertile)	−0.279 (0.111) ^b^	0.005 (0.006)	−0.097 (0.195)	0.007 (0.010)	1.22 (0.91, 1.64)	0.85 (0.65, 1.10)
Taurine (per one tertile)	−0.066 (0.100)	−0.008 (0.005)	−0.011 (0.174)	−0.009 (0.009)	1.41 (1.08, 1.84) ^b^	1.11 (0.87, 1.42)
tGSH (per one tertile)	0 (0.101)	−0.008 (0.005)	0.122 (0.176)	0.028 (0.009) ^b,c^	0.92 (0.71, 1.19)	1.02 (0.80, 1.30)

DII: dietary inflammatory index; OR: odds ratio; CI: confidence interval; SE: standardized error. tHcy: total homocysteine; tCys; total cysteine; tGSH: total glutathione. 1423 men and 1533 women are available for serum amino acids assessment. ^a^ in fully adjusted model B: adjusted for baseline age, corresponding measurement, body mass index, estimated glomerular filtration rate, current smoking, physical activity level, previous fracture, hypertension, diabetes, chronic obstructive lung disease, cardiovascular disease, rheumatoid arthritis, nonsteroidal anti-inflammatory agent use, and osteoporosis medication. ^b^ indicates for significant value (*p* < 0.05). ^c^ indicates for significant value (*p* < 0.004) for amino acids. ^d^ in those without sarcopenia at baseline. ^e^ in sub-fully adjusted model C: adjusted for baseline age, corresponding measurement, estimated glomerular filtration rate, current smoking, physical activity level, previous fracture, hypertension, diabetes, chronic obstructive lung disease, cardiovascular disease, rheumatoid arthritis, nonsteroidal anti-inflammatory agent use, and osteoporosis medication.

## Data Availability

The data that support the findings of this study are available from the corresponding author upon reasonable request.

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
