# Peer review of "The Association of Circulating Amino Acids and Dietary Inflammatory Potential with Muscle Health in Chinese Community-Dwelling Older People"

_nutrients, 2022, doi:10.3390/nu14122471_

Round 1

Reviewer 1 Report

Abstract
It must remove the abbreviations from the abstract and especially those that are not defined as DII
Introduction
The introduction should be reviewed, which is very generalized for the role of some amino acids such as leucine, where the amount of research reveals its role in the integrity of the muscle and there are even intake recommendations for its regeneration (see Recommendations from the ESPEN Expert Group)
Material and Method
At no time in the study do the authors speak of sarcopenia when they obtain all the variables that would allow them to make a diagnosis.

In material and method, it is not clear that only an evaluation of the diet is made at the beginning. This aspect is not clear until the limitations of the study are reached.

Results

Having such a large sample, it would be of interest to reveal the prevalence of sarcopenia in men and women
Although the abbreviations are defined in the text, in the table footers they must be defined again. These aspects make it difficult for a non-expert reader. Alphabetize the abbreviations at the foot of the table
In the results it is not clear whether a pro-inflammatory diet was taken in the sample or only analyzed to know-how is the diet. If a study is carried out for 4 years with an evaluation at the beginning and another at the end, it is expected that the population will modify their diet? These aspects are not clear when reading the article.
In the results, it is not clear whether a pro-inflammatory diet was taken in the sample. If a study is carried out for 4 years with an evaluation at the beginning and another at the end, it is expected that the population will modify their diet? These aspects are not clear when reading the article.

It has not been indicated which are the cut-off points to consider low mass or low strength and what happens after 4 years. In older people, the important thing is the functionality of the muscle.

Reviewer 2 Report

This is an interesting and highly significant study that looks at the association of circulating amino acids and dietary inflammatory potential with muscle health in Chinese community-dwelling older people. The authors collected demographic data, performed physical assessment and analyzed the serum from 2000 Chinese older men and 2000 Chinese older women. The authors also conducted a follow-up study four years later. The authors reported that the older men and women consumed a pro-inflammatory diet that contributes to high DII and weaker muscle strength. Overall, the manuscript is easy to follow and understand. Some of my minor comments are as follow:

1.    Abstract: What is DII? Please include its full word in the abstract.

2.    It is important to list the 280 (or maybe those commonly consumed) food items (in the supplemental) along with the compositions that may contribute to high DII. These parameters then should correlate to Table 1.

3.    Controls (people who do not consumer pro-inflammatory diet) should be included. These controls will be important when comparing the healthier food and pro-inflammatory diet.

Round 2

Reviewer 1 Report

The authors have significantly improved the article